# Thermo-Compression Bonding of Cu/SnAg Pillar Bumps with Electroless Palladium Immersion Gold (EPIG) Surface Finish

**DOI:** 10.3390/ma16041739

**Published:** 2023-02-20

**Authors:** So-Yeon Jun, Jung-Hwan Bang, Min-Su Kim, Deok-Gon Han, Tae-Young Lee, Sehoon Yoo

**Affiliations:** 1Advanced Joining & Additive Manufacturing R&D Department, Korea Institute of Industrial Technology, Incheon 21999, Republic of Korea; 2MK Chem. & Tech., Ansan, Kyunggido 15434, Republic of Korea

**Keywords:** thermo-compression bonding (TCB), electroless palladium immersion gold (EPIG), electroless nickel electroless palladium immersion gold (ENEPIG)

## Abstract

Thermo-compression bonding (TCB) properties of Cu/SnAg pillar bumps on electroless palladium immersion gold (EPIG) were evaluated in this study. A test chip with Cu/SnAg pillar bumps was bonded on the surface-finished Cu pads with the TCB method. The surface roughness of the EPIG was 82 nm, which was 1.6 times higher than that of the ENEPIG surface finish because the EPIG was so thin that it could not flatten rough bare Cu pads. From the cross-sectional SEM micrographs, the filler trapping of the TC-bonded EPIG was much higher than that of the ENEPIG sample. The high filler trapping of the EPIG sample was due to the high surface roughness of the EPIG surface finish. The contact resistance increased as the thermal cycle time increased. The increase of the contact resistance with 1500 cycles of the thermal cycle test was 26% higher for the EPIG sample than for the ENEPIG sample.

## 1. Introduction

Recent improvements in performance and node size decrease in semiconductors lead to a lowering in the bump pitch of the semiconductor package [1,2,3]. The fine-pitch semiconductor packages have brought about a huge change in the flip-chip bonding process and materials. A thermo-compression bonding (TCB) with non-conductive adhesive (NCA) is increasingly replacing traditional mass flow C4 processes [4,5,6]. Substrate pads to which the chip bumps are bonded are also finer, and therefore, the thickness of the surface finish plated on the substrate pad should be thinner than at least half of the pad pitch. An electroless nickel electroless palladium immersion gold (ENEPIG) surface finish has been widely used for highly reliable mobile devices due to its long-term shelf life and is good for multiple heat treatments [7,8,9]. In addition, the palladium layer in the ENEPIG prevents the corrosion of the Ni–P layer, which enhances solder wettability [10,11]. However, the total thickness of the ENEPIG surface finish is 4~5 μm, which causes concerns about bridging adjacent pads at the fine pitch pads. Moreover, the electroless nickel in the ENEPIG surface finish is ferromagnetic, which causes an adverse effect that attenuates a signal in the 5G application [12,13]. Recently, Ni-less surface finishes, electroless palladium immersion gold (EPIG), have been developed to overcome the signal attenuation and to apply the fine pitch substrate [12]. The thickness of the EPIG surface finish was lower than 0.5 μm. In addition, The Ni-less surface finish showed 45% lower signal loss at 50 GHz than conventional Ni-included surface finishes such as ENIG since there was no ferromagnetic Ni layer in EPIG [13]. Therefore, the Ni-less surface finishes are expected to be widely applied in 5G/6G communications. 

Compared with the conventional ENEPIG, the TCB joint characteristics of the EPIG surface finishes with Cu pillar bumps have rarely been reported. Therefore, the TCB joint characteristics of the Cu pillar joints on the EPIG need to be investigated. In this study, TCB with NCA was used for the Cu pillar bump joints with diameters of 30 μm. The interfacial reactions and electrical contract resistance of the joints were evaluated to understand the effect of surface finishes on the joint properties.

## 2. Materials and Methods

A Si test chip with Cu/Sn-3.5wt%Ag (Cu/SnAg) pillar bumps and a bismaleimide triazine (BT) substrate were used in this study, as shown in Figure 1. The Cu/Sn3.5Ag means a bump structure with Sn-3.5wt%Ag solder on the Cu pillar bump. The size of the test chip and the substrate were 4.4 × 4.4 mm^2^ and 10 × 10 mm^2^, respectively. Cu/SnAg pillar bumps were formed on the test chip and are shown in Figure 2. The diameter of the Cu/SnAg pillar bumps was 30 μm. The heights of the Cu pillar and SnAg cap of the pillar bumps were 15 μm and 10 μm, respectively. The total number of pillar bumps on a test chip was 735. 

Before the electroless plating process of the EPIG, a pre-treatment was performed to clean and strengthen the adhesion of the surface finish. First, the pre-treatment consisted of degreasing, soft-etching, and catalyst treatments. The degreasing was performed for the purpose of removing contaminants on the substrate, and the degreasing conditions were 45 °C for 5 min. Next, soft etching was performed to provide strong adhesion to the solder resist by forming surface roughness on the Cu pad. Soft etchant was produced by dissolving 100 g of a soft etching solution (Oxone PS-16, Dupont, Wilmington, NC, USA) with 15 mL of 95% sulfuric acid. The soft etching condition was 27 °C for 70 s. Finally, the surface activation process was performed by dipping the substrate in palladium catalyst solution (ICP Accera H series, MK Chem. & Tech., Ansan, Republic of Korea) at 27 °C for 1 min. 

For the EPIG surface finish, electroless palladium and substitutional gold plating were carried out at 65 °C and 83 °C, respectively. The electroless palladium plating and substitution gold plating solutions were obtained from MK Chem & Tech (Neozen Pd-P, Flash Gold IG-10, respectively). As a control sample, ENEPIG surface finish was also fabricated. The electroless nickel plating solution was obtained from MK Chem. & Tech. (Neozen MP-K Series), and the plating temperature was 83 °C. The Pd and Au plating processes were the same as those of the EPIG surface finish. The schematic diagrams of the EPIG and the ENEPIG surface finishes are shown in Figure 3. The thicknesses of the palladium and gold layers in EPIG were 0.1 and 0.1 μm, and the thicknesses of the nickel, palladium, and gold layers in ENEPIG were 5, 0.1, and 0.1 μm, respectively. A daisy chain and Kelvin structure electrodes were formed on the substrate to measure electrical resistance and contact resistance.

The Cu/SnAg pillar bumps of the test chip and the surface-finished Cu pad of the test substrate were bonded to each other through a TCB technique with a flip-chip bonder (NM-SB50A, Panasonic, Osaka, Japan). Before the TCB process, non-conductive adhesive (NCA) was dispensed to the substrate. The NCA formulation method can be found elsewhere [4]. The NCA included 55 wt% silica fillers with a size of 800 nm. After dispensing the NCA, the test chip was aligned to the NCA-dispensed substrate and bonded with a pressure of 20 N. The schematic of the bonding process and temperature profile is shown in Figure 4, and the TCB sample is shown in Figure 5. The peak temperature of the TCB was 250 °C, and the hold time at the peak temperature was 5 s. Following the TCB process, the NCA was post-cured at 160 °C for 60 s in an oven. 

The roughness of the EPIG and ENEPIG surface finishes and the joint Cu/SnAg pillar bumps on the Cu pads were observed with scanning electron microscopy (SEM; Inspect F, FEI Co., Hillsboro, OR, USA), and the chemical composition of the solder joint interface was observed with energy-dispersive X-ray spectroscopy (EDS; Superdry II, Therom Fisher Scientific, Waltham, USA). The surface morphology of the surface finish was observed with atomic force microscopy (AFM; XE7, Park System, Suwon, Republic of Korea). To evaluate the reliability of the surface finishes, a thermal cycling test was conducted with a thermal cycling tester (VCS 7027-15, Votsch, Balingen, Germany) for 1500 cycles. The temperature range of the one cycle was from −55 °C to 125 °C. Additionally, the contact resistance was measured during the thermal cycle test. The electrical contact resistance of a Cu/SnAg pillar bump joint was evaluated with a four-point probe (Everbeing, C-6) [14]. A Kelvin structure, which was a specially designed pattern for the four-probe measurement, was fabricated on the chip and to measure the contact resistance, as shown in Figure 6. To measure the contact resistance, an electric current was applied through the electrodes marked I1 and I2, and voltage was measured between V1 and V2. We measured the contact resistance of 16 bumps and obtained the average and standard deviation.

## 3. Results and Discussion

The surfaces of the bare Cu, EPIG, and ENEPIG surface-finished pads were observed with SEM and shown in Figure 7. From the enlarged top view of the surface finish, the surface of the bare Cu pad was much rougher than that of the surface-finished pads. Such a rough surface of the bare Cu pad was due to the soft etching process. A soft etching is a process that removes Cu oxide and provides surface irregularity to improve adhesion between the Cu pad and solder resist [15]. For the EPIG sample, the surface roughness was slightly decreased compared to the bare Cu pad. The ENEPIG surface finish was relatively smooth compared to bare Cu and EPIG. This was because the rough surface of bare Cu was covered with a 5 μm thick electroless nickel layer. 

The surface topographies and the line profiles of the soft-etched bare Cu pad and surface-finished pads were observed with AFM and shown in Figure 8. From the surface topographies, the surfaces of the EPIG and ENEPIG finishes were smoother than the soft-etched Cu pad, which indicated that the surface finish process clearly lowered the surface roughness of the Cu pad. From the line profile measurement, the root-mean-square (RMS) roughness values of the bare Cu pad, EPIG, and ENEPIG finishes were approximately 90, 82, and 50 nm, respectively, which was evaluated from the line profile in Figure 8.

The surface roughness of EPIG was 1.6 times higher than that of ENEPIG. The surface roughness differences among the surface finishes can be explained with a schematic (Figure 9). Before the surface finish plating process, the Cu pad had a high surface roughness due to the soft etching process. The high surface roughness of these Cu pads gradually decreased as the surface finish layer was coated on the Cu pad. As the plating thickness increased, the rough surface became smoother because the surface finishes covered the Cu pad. The EPIG had high surface roughness since the EPIG had a nanoscale thin layer, which was too thin to cover the rough Cu pad. On the other hand, the surface finish of ENEPIG exhibited low roughness because the Cu pad was significantly covered with a thick (5 μm) nickel layer. Therefore, the surface roughness of the Cu pad was distinctively affected by the thickness of the surface finish. 

Figure 10 shows TC-bonded Cu/SnAg pillar bumps on surface-finished Cu pads. An intermetallic compound (IMC) was formed between solder and surface finishes. For the EPIG surface finish, the IMC phases were (Cu,Pd)_6_Sn_5_ and (Pd,Cu)Sn_4_, as observed by EDS. On the other hand, the IMC of the ENEPIG surface finish was Ni_3_Sn_4_, as reported by many research groups [7,8,9,16,17]. In addition, the trapped NCA filler was observed in the lower part of the solder (see Figure 10). NCA resin was also observed near the trapped fillers. The NCA pillar was silica with an average diameter of 800 nm. The role of the silica filler is to lower the coefficient of thermal expansion (CTE) of the NCA, which ensures the reliability of the TCB joints [5]. During the TCB process, the downward force of the bump squeezed out most of the fillers, but few fillers remained at the solder joint interface. Such filler trapping adversely affects the reliability of the pillar bump joints [5,16]. When the filler is trapped at the solder joint interface, the NCA resin is also trapped. As a result, the trapped NCA resin can cause cracking and delamination during the thermal cycle.

Generally, filler traps in the pillar bump joints substantially led to a contact resistance increase because the contact area between a bump and a Cu pad decreased [18,19,20,21]. To evaluate the degree of the filler trapping of the Cu/SnAg pillar bump joints quantitatively, the contact resistance was measured and shown in Figure 11. The contact resistances of EPIG and ENEPIG were approximately 4.2 and 3.5 mΩ, respectively. The average contact resistance of ENEPIG was lower than that of EPIG. The higher contact resistance of the EPIG sample indicated that the EPIG had a higher degree of filler trapping at the pillar bump joint. Lee et al. [18] reported that the surface roughness of the bump significantly affects the amount of filler trapping. Park et al. [20] also reported that a smooth Cu pad sample showed lower filler trapping than a rough Cu pad sample for low-temperature TCB. In this study, the roughness of EPIG was higher than that of ENEPIG surface finish, and the high roughness of the EPIG sample increased the filler trap and the contact resistance. The rough surfaces of the EPIG surface finishes had more dimples and valleys than that of the ENEPIG surface finish. These dimples and valleys finally become the entrapment sites for the filler and resin during the bonding process [22]. 

Figure 12 shows the contact resistance change of each surface finish sample during a thermal cycle test. At cycle 0, the contact resistance difference between EPIG and ENEPIG was 0.7 mΩ. As the cycles increased, the difference in contact resistance between the EPIG and ENEPIG samples gradually increased. At cycle 1500, the contact resistances of the EPIG and ENEPIG samples were approximately 75.8 and 62.3 mΩ, respectively. The contact resistance difference between the EPIG and the ENEPIG sample was 13.5 mΩ. The lower contact resistance of the ENEPIG sample than that of the EPIG sample showed that the ENEPIG sample had fewer defects than the EPIG sample. Since all samples were bonded under the same conditions, the difference in contact resistance came from the degree of the filler trapping. The contact resistance increased with the increasing thermal cycle due to the expansion of the resin included with the trapped fillers. Thus far, we have reported the TC bonding properties of the EPIG surface finish with Cu/SnAg pillar bumps. 

The surface roughness differences affected contact resistance and TC reliability due to NCA filler trapping. Therefore, lowering the surface roughness of the EPIG surface finish should be required for the adoption of the Ni-less surface finish for TC bonding.

## 4. Conclusions

In this study, the TC bonding properties of Cu/SnAg pillar bumps on EPIG were evaluated with the electrical contact resistance. The following conclusions have been drawn on the basis of these experiments:·Filler trapping was observed at the Cu/SnAg pillar joints on the surface finishes. The EPIG samples had more filler trapping than the ENEPIG sample. ·The EPIG sample had higher contact resistance than the ENEPIG sample. The contact resistance difference came from the degree of the filler trapping. ·The EPIG sample had higher contact resistance with the thermal cycle than the ENEPIG sample. The contact resistance increase was due to the expansion of the trapped NCA resin. 

## Figures and Tables

**Figure 1 materials-16-01739-f001:**
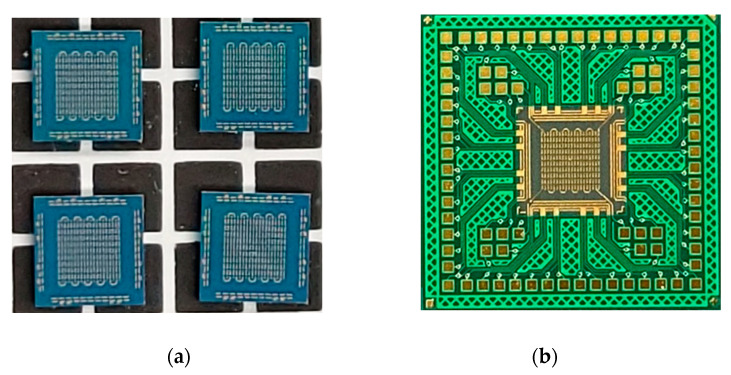
(**a**) Test chip and (**b**) test substrate in this study.

**Figure 2 materials-16-01739-f002:**
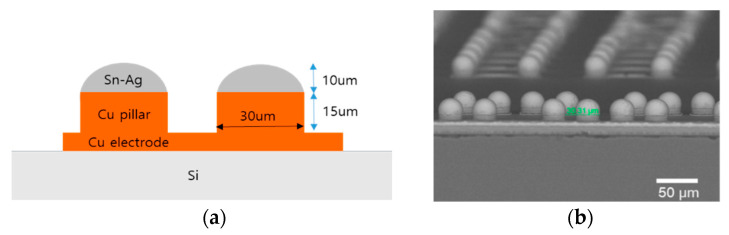
(**a**) Schematic and (**b**) SEM micrograph of Cu/SnAg pillar bump.

**Figure 3 materials-16-01739-f003:**
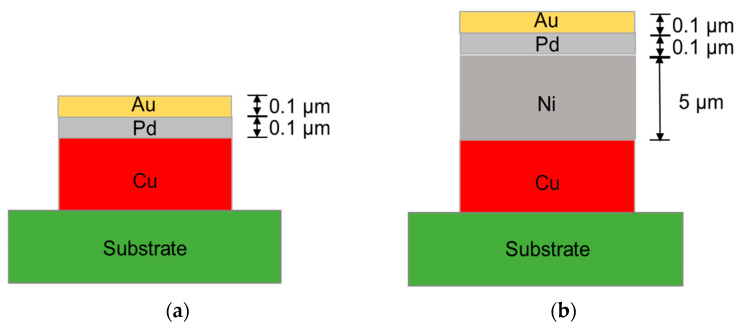
Schematics of the surface finishes; (**a**) electroless palladium immersion gold (EPIG) and (**b**) electroless nickel electroless palladium immersion gold (ENEPIG).

**Figure 4 materials-16-01739-f004:**
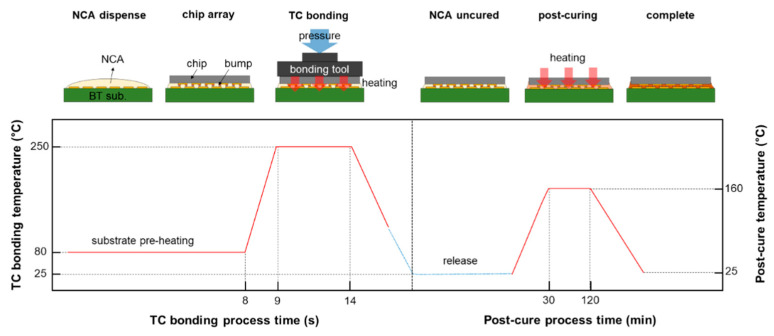
Schematic of bonding process with temperature profile of the TCB process and the post-cure treatment.

**Figure 5 materials-16-01739-f005:**
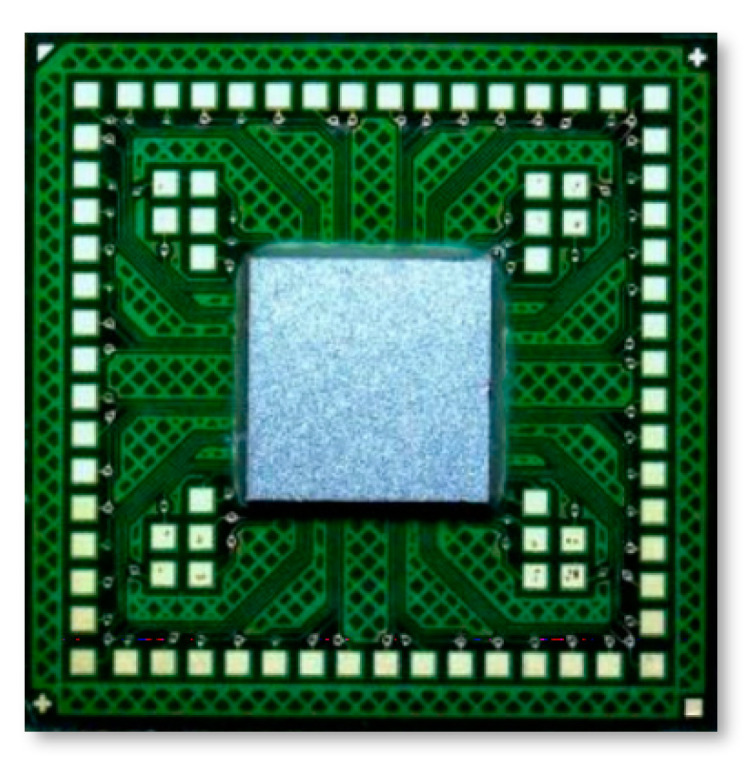
Optical micrograph of thermo-compression (TC)-bonded test sample.

**Figure 6 materials-16-01739-f006:**
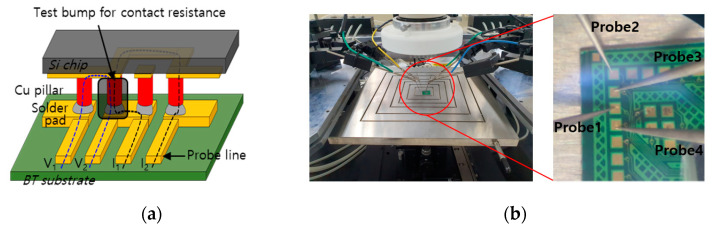
(**a**) Schematic of Kelvin structure for contact resistance measurement and (**b**) 4-point probe for the contact resistance measurement.

**Figure 7 materials-16-01739-f007:**
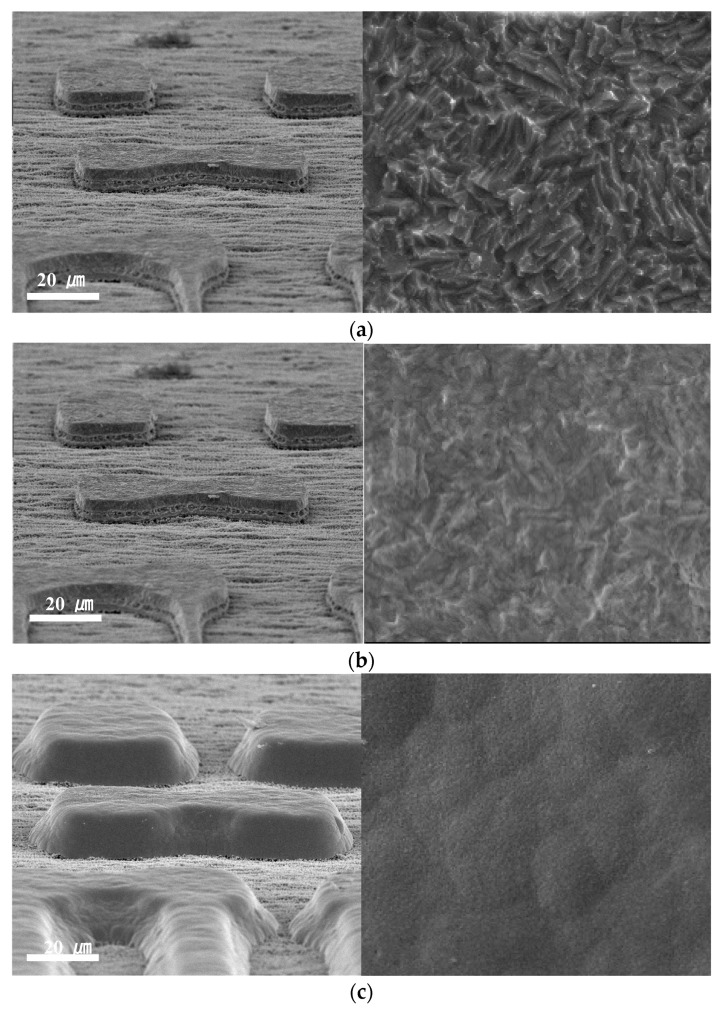
Scanning electron micrographs of (**a**) bare Cu, (**b**) EPIG, and (**c**) ENEPIG. Left images show tiled view, and right images show the enlarged top view of the surface finishes.

**Figure 8 materials-16-01739-f008:**
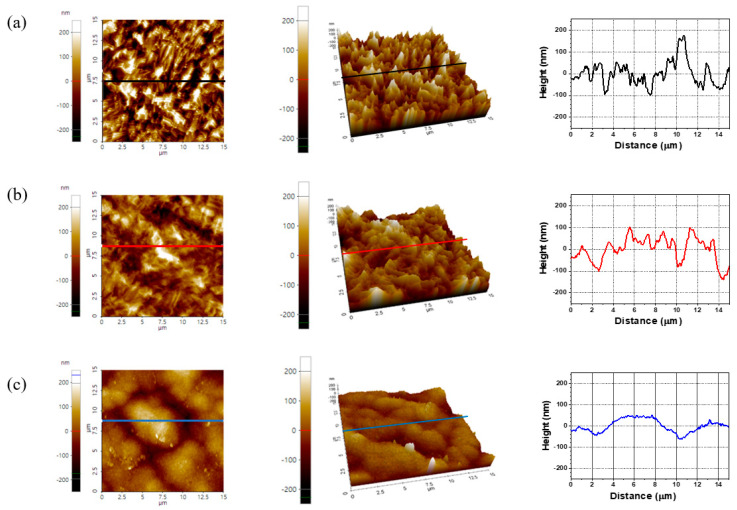
AFM surface topography images of (**a**) bare Cu, (**b**) EPIG, and (**c**) ENEPIG. Left and middle images show 2D and 3D views of AFM, respectively, and right images show line profiles indicated by lines in the 2D and 3D AFM images.

**Figure 9 materials-16-01739-f009:**
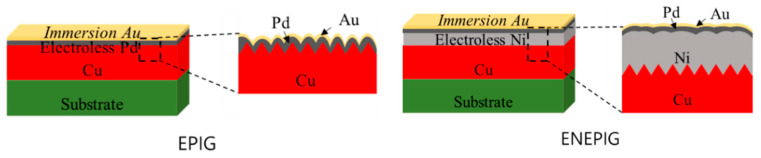
Schematics that explain the surface roughness difference between EPIG and ENEPIG surface finishes.

**Figure 10 materials-16-01739-f010:**
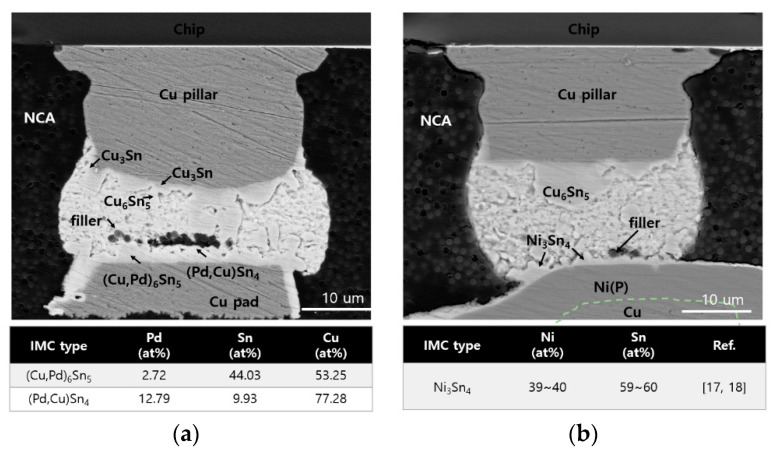
Cross-sectional SEM micrographs and EDS of Cu/SnAg pillar bumps on (**a**) EPIG and (**b**) ENEPIG surface finishes.

**Figure 11 materials-16-01739-f011:**
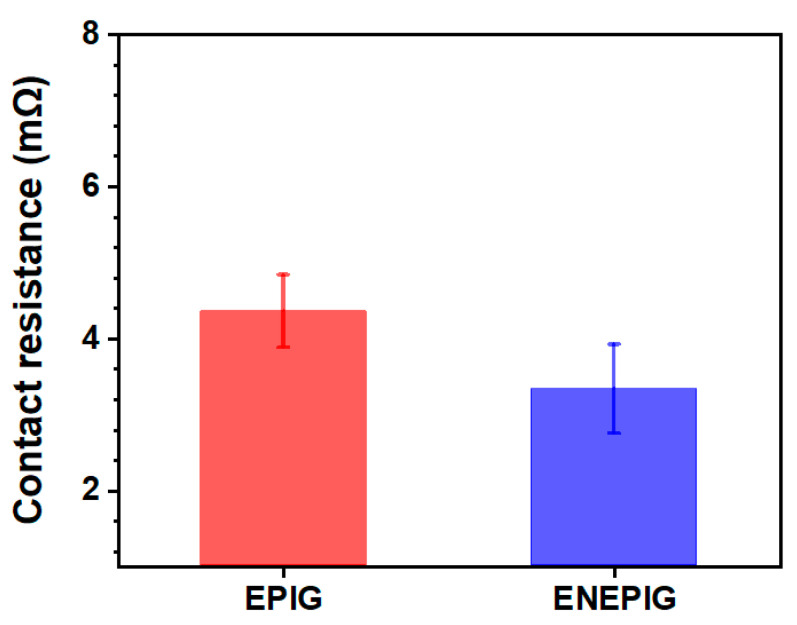
Contact resistance of Cu/SnAg pillar bump joints on surface finishes, EPIG and ENEPIG.

**Figure 12 materials-16-01739-f012:**
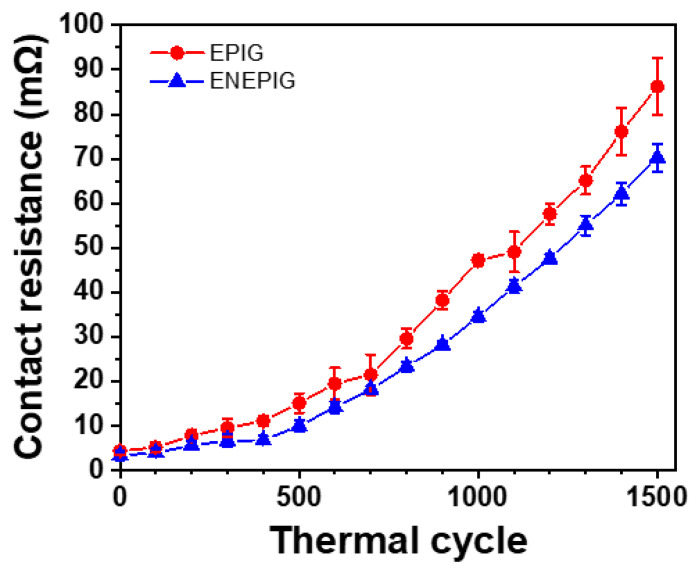
Contact resistance of Cu/SnAg pillar bump joints on EPIG and ENEPIG surface finishes with thermal cycling test.

## Data Availability

All data are presented within the manuscript.

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
