# Peer review of "Thermo-Compression Bonding of Cu/SnAg Pillar Bumps with Electroless Palladium Immersion Gold (EPIG) Surface Finish"

_materials, 2023, doi:10.3390/ma16041739_

Round 1

Reviewer 1 Report

Dear authors, 

1. Please revise line 49-50

2. Themocompression or themos compression? please check for consistency

3. SEM, FEI Inspection F..or Inspect F?

4. Line 164-165: (Cu, Ni)6Sn5 or Ni3Sn4 present for fig 10? if EDX data can be included it would be great. The same goes if EDX data can be provided to verify the trapped NCA filler.

5. If shear/tensile test + microstructure failure analysis can be added to support the loss of mechanical properties due to the NCA filler trapped, it would give great insight. 

Reviewer 2 Report

The manuscript focuses on investigation of thermocompression bonding of Cu/SnAg pillar bumps on electroless palladium immersion gold. However, the abstract of the manuscript has to be extended, explaining why the surface roughness of the electroless palladium immersion gold was so thin that it could not flatten rough bare Cu pads. It looks that this conclusion or result is trivial and there is no need to mention it in the abstract. For scientific papers the statements ‘…much higher…’, ‘…so thin…’ (line 16), ‘…higher…’ (line 18), ‘…much higher…’ (line 19), ‘…due to high surface roughness…’ are inappropriate. The abstract has to be reworded making it sound scientifically and/or giving some numeric values and letting the readers compare themselves which one is ‘higher’ and which one is ‘lower’.

Lines 45-47: Authors mention that <…> the Ni-less surface finish showed much lower signal loss than the conventional Ni-included surface finishes<…> giving no info on that what rate that ‘much lower’ is.

Line 56: Authors have to explain the meaning of symbols ‘Cu/Sn-3.5wt%Ag’, ‘Cu/Sn3.5Ag‘;’Cu6n5’

Lines 66-67: the regime of degreasing at 45oC for 5 min is not sufficiently clear in terms of whether it is optimal in that case proposed in the manuscript or not. The same applies to the regime of soft etching and/or dipping of the substrate in palladium catalyst solution (see lines 70-71). Same in lines 88-89, 110-111. 

Not sufficiently clear function and measurements units on ordinate axes in Figure 8 (on the right). Too-small symbols that demonstrate surface roughness (on the left). If printed, they will hardly be seen.      

Lines 182-183. Authors gave results of contact resistance measurements, but no information about a way/method that has been applied for the contact resistance measurements and/or what are error bars of these measurements. Same question related to the contact resistance of the EPIG and ENEPIG samples in lines 198-199. Because contact resistance is a function of the contact area between the contact material and the specimen, the estimation of the contact area of the contact produced by the thermocompression bonding method would also be useful to better understand the physics of difference in contact resistances in the case of EPIG and ENEPIG samples. I think it would make some sense for the better scientific soundness of the manuscript.      

I think that the sentence: ‘Since all samples were bonded with same conditions, the contact resistance difference came from the degree of the filler trapping’ is of bad style with mistaken English. It must be replaced with the following: ‘Since all samples were bonded under the same conditions, the difference in contact resistance came from the degree of filler trapping.’

The title of the function on the abscissa axis of Figure 12 is not sufficiently clear. Especially if the word ‘cycle’ is used in the singular form.

The sentences in lines 215-218 state a trivial fact. I propose removing it from part of the Conclusions.

Misprinting errors:

In line 18.  ‘Contact resistance of the thermocompression(TC)-bonded EPIG sample was higher than the that of the ENEPIG sample.’ Has to be corrected. I think it should be, ‘The contact of the thermocompression(TC)-bonded EPIG sample was higher than that of the ENEPIG sample.’

In line 169. Wrong expression stating that ‘the coefficient of temperature expansion’ exists. I think the temperature cannot expand.

Reviewer 3 Report

The work on "Thermocompression bonding of Cu/SnAg pillar bumps with electroless palladium immersion gold (EPIG) surface finish", is appreciable. But, below mentioned comments must be addressed before possible recommendations for the publication.

Comments:

1. During the TCB process,  why the fillers which are remained at the solder joint interface,  affects the reliability of the pillar bump joints? Explain it.

2. Why the number of trapped fillers was higher for the EPIG samples  than ENEPIG sample?

3. What about the leakage around the solder joint interface? Comment on this.
